# Reduced Rate of Anemia after Transcatheter Aortic Valve Replacement

**DOI:** 10.3390/jcm13185606

**Published:** 2024-09-21

**Authors:** Amnon Eitan, Hussein Sliman, Barak Zafrir, Keren Zissman, Moshe Y. Flugelman, Ronen Jaffe

**Affiliations:** 1Department of Cardiovascular Medicine, Lady Davis Carmel Medical Center, 7 Michal Street, Haifa 34632, Israel; 2Rappaport Faculty of Medicine, Technion-Israel Institute of Technology, Haifa 3109601, Israel

**Keywords:** anemia, angiodysplasia, von Willebrand factor, renal function

## Abstract

**Background/Objectives:** We sought to evaluate changes in hemoglobin level and renal function in patients 5–12 months after transcatheter aortic valve replacement (TAVR), and to examine possible relationships between these changes. Anemia is common in older people with severe aortic stenosis (AS). The two most common etiologies for anemia in this population are iron deficiency due to gastrointestinal blood loss and renal failure. Angiodysplasia in the gastrointestinal system is a feature of AS syndrome. **Methods:** We collected clinical data, including hemoglobin level and renal function before and 5–12 months after TAVR in 315 consecutive patients. To examine whether calculated clinical predictors such as EuroScore 2 are associated with the persistence of anemia after TAVR, we performed multivariable correlation analysis with post-TAVR anemia as the dependent variable. **Results:** The mean hemoglobin level increased significantly (from 11.76 to 12.16 g/dL, *p* < 0.0001) 5–12 months after TAVR, and the number of patients with anemia decreased significantly (from 67.5% to 53.9%, *p* < 0.0001). At 5–12 months following TAVR, a small reduction in estimated glomerular filtration rate was observed (from 60.05 ± 24.1 to 58.30 ± 24.50 mL/min, *p* = 0.024). The multivariable correlation analysis did not identify clinical predictors of persistent anemia. **Conclusions:** A significant increase in hemoglobin was observed 5–12 months after TAVR, despite a reduction in renal function. Our findings imply that gastrointestinal blood loss, which occurs in patients with severe AS, is significantly reduced following TAVR.

## 1. Introduction

Transcatheter aortic valve replacement (TAVR) is indicated in patients with severe aortic stenosis (AS) who are at moderate or high surgical risk [1]. TAVR is usually performed by transfemoral approach. Candidates for TAVR are usually older people with multiple comorbid conditions, who are often anemic. Iron deficiency presents in more than half the candidates for TAVR, and renal dysfunction is the second most prevalent cause of anemia in this population [2,3,4]. Iron deficiency usually results from gastrointestinal blood loss. Heyde’s syndrome is a distinct condition in patients with severe AS, which is characterized by reduced serum levels of von Willebrand factor and the presence of gastrointestinal angiodysplasia that is prone to bleeding [5]. Several case reports have described the resolution of angiodysplasia and of iron loss following TAVR [5,6,7,8]. Renal dysfunction presents with a high prevalence among patients who undergo TAVR, is associated with worse procedural outcomes [9,10,11,12], and may be an additional cause of anemia. Scarce data are available regarding changes in hemoglobin level, the prevalence of anemia, and renal function after TAVR [13]. We sought to evaluate changes in hemoglobin level and renal function in patients 5–12 months after TAVR, and to examine possible relations between these changes.

## 2. Methods

We reviewed our prospective institutional TAVR registry and collected baseline clinical, demographic, and procedural characteristics, as well as in-hospital outcomes. Follow-up was performed at the valve clinic and via the national electronic medical database system (Horizon).

Safety outcomes during the first 30 days were evaluated according to the Valve Academic Research Consortium-2 (VARC-2) criteria [14]. Hemoglobin level, red blood cell indexes, and creatinine level were recorded 5–12 months after the procedures. Anemia was defined according to World Health Organization definitions as hemoglobin <13 g/dL in men and <12 g/dL in women [15]. Estimated glomerular filtration rate (eGFR) was calculated using the Cockcroft–Gault formula. Chronic kidney disease (CKD) stages were defined according to the Kidney Disease: Improving Global Outcomes (KDIGO) organization guidelines [16]. We evaluated a correlation between changes in renal function and in hemoglobin level. The cohort comprises the patients for whom complete data were available.

To examine whether calculated clinical predictors such as Euroscore 2 are associated with persistence of anemia after TAVR, we performed multivariable correlation analysis with post-TAVR anemia as the dependent variable.

The study protocol conforms to the ethical guidelines of the 1975 Declaration of Helsinki, as reflected in a priori approval by the institution’s human research committee. 

### Statistical Analysis

Continuous variables are presented as means and standard deviations, and categorical variables as frequencies and percentages for each group. For comparison of continuous variables, Student’s *t*-test was used. Chi-square and Fisher’s exact tests were used to compare differences between proportions. Linear regression was conducted to evaluate relations between variables. Statistical analysis was performed with SPSS v.27 software.

## 3. Results

Between 1 January 2016 and 31 August 2019, 374 TAVR procedures were conducted at our center. Patients after TAVR with no other anti-thrombotic indication are treated with Aspirin as a life-long therapy. At the referred-to years, the practice at our center was to add Clopidogrel for 3–6 months. At 5–12 months after TAVR, hemoglobin and renal function data were available for 315 patients; their baseline characteristics are presented in Table 1. The mean age was 80 years; 53% were female. Hypertension, coronary artery disease, diabetes mellitus, and atrial fibrillation were common (88%, 50%, 38%, and 29%, respectively).

Transfemoral vascular access was used in all the patients, except for two in whom the trans-axillary approach was used. Thirty-day safety outcomes occurred in 7% and acute kidney injury in 2%. During the 5–12 months following TAVR, the mean hemoglobin and hematocrit levels significantly increased from 11.79 ± 1.65 to 12.18 ± 1.64 g/dL and from 36.50 ± 5.74 to 37.97 ± 4.98 percent, respectively (*p* < 0.0001 for both). The prevalence of anemia decreased from 66.7% to 53.0% (*p* < 0.0001). The mean hemoglobin level increased significantly among patients with anemia, from 10.94 to 11.66 g/dL (*p* < 0.0001). This compares with a decrease in mean hemoglobin level that was statistically significant though of small magnitude among patients without anemia, from 13.50 to 13.20 g/dL (*p* = 0.025). No change in mean corpuscular volume was observed, from 88.85 ± 6.82 to 89.52 ± 7.00 (*p* = 0.063). The changes in hemoglobin following TAVR did not differ significantly according to gender, valve type, or the presence of diabetes mellitus or hypertension at baseline (Table 2).

At 5–12 months after TAVR, a small reduction in mean eGFR was observed, from 60.05 ± 24.1 to 58.30 ± 24.50 mL/min (*p* = 0.024). The changes in eGFR were not associated with the presence of anemia, diabetes mellitus, or hypertension; or according to gender or valve type (Table 3). Following TAVR, 234/315 (74%) of the patients remained in the same baseline CKD stage or improved; while for 81 patients (26%), the CKD stage deteriorated. Univariate analysis showed no relationships of demographics and EuroScore 2 to a change in hemoglobin level after TAVR (Table 4). In the linear regression model, neither eGFR at baseline nor the change in eGFR predicted a change in hemoglobin level. Statistically significant associations were also not found in the multivariable analysis (Table 5).

## 4. Discussion

We found a statistically significant increase in mean hemoglobin level 5–12 months after TAVR, and a decreased rate of anemia. The increase in hemoglobin level occurred despite a small but significant reduction in renal function at 5–12 months following TAVR.

Gastrointestinal blood loss in AS has been attributed to bleeding from high shear stress in blood vessels, such as in angiodysplasia, in the presence of a quantitative loss of the highest molecular weight von Willebrand multimers (type 2A von Willebrand syndrome) or so-called Heyde’s syndrome [5,17]. Casper et al. showed an increase in von Willebrand factor and a reduced rate of anemia following TAVR [18]. However, angiodysplasia may also conceivably result from changed patterns of blood flow, with blunting of the pulsatile flow, as occurs in AS and in patients with a left ventricular assist device [19,20,21]. Blunting of the arterial pulse wave and the presence of abnormal arterial structures probably cause the change in von Willebrand homeostasis, as was shown by Casper et al. [18] and by Akutagawa et al. [22]. The latter case report described the persistence of angiodysplasia after TAVR, but without bleeding or increased von Willebrand factor. This scenario might explain the greater tendency to bleed among patients with AS who are treated with dual anti-platelet therapy after myocardial infarction [23]. A recent study by Goltstein et al. reported a significant reduction in gastrointestinal bleeding, after TAVR, in the majority of patients with Heyde syndrome. In the first year following TAVR in their study, gastrointestinal bleeding ceased in 46 of 70 patients (62% [95% CI, 50–74%]) and hemoglobin levels increased from 10.3 (95% CI, 10.0–10.8) to 11.3 (95% CI, 10.8–11.6) g/dL (*p* = 0.007) [24]. Also, a meta-analysis by Goltstein et al. found the pooled proportion of Heyde patients with gastrointestinal bleeding cessation after TAVR to be 73% (62–81%) [25]. Yashige et al. [26] published, in a brief report, the significant reduction in the number and size of the angiodysplastic lesions in patients with endoscope-diagnosed Hyde syndrome after TAVR. No patient had active bleeding following TAVR [26].

We did not measure levels of von Willebrand factor before or after TAVR nor did we have endoscopic data. However, in light of the deterioration in renal function in our patients, reduced iron loss remains the most probable explanation for the lower rate of anemia. Our data may be more generalized and relevant to patients with no overt evidence of Hyde syndrome, as blood loss may progress very slowly.

The change in hemoglobin level did not differ between patients with and without common comorbidities. Multivariable analysis also did not identify any clinical feature at baseline, including the change in eGFR, as a predictor of change in hemoglobin. As other reasons for anemia, such as renal failure, were not found to be predictors of change in anemia, our findings implicate reduced loss of blood through the gastrointestinal system after the relief of AS. Interestingly, in our cohort, hemoglobin was reduced at 5–12 months in patients without anemia prior to the TAVR procedure. We have no explanation for this finding other than the use of anti-platelet therapy and blood loss ascribed to reduced platelet activity. 

Our findings concur with those of De Backer at al., who demonstrated recovery from anemia in about 40% of patients with this condition prior to TAVR [13]. Elsewhere, a low rate of deterioration of renal function was reported at one week after the TAVR procedure [27]. A multicenter study reported similar findings 30 days after TAVR among survivors of the procedure [28]. Interestingly, kidney function recovery after TAVR was reported as less common among patients with baseline anemia and diabetes [29]. Since severe AS compromises cardiac output, it may affect renal perfusion and thus contribute to renal dysfunction. Accordingly, long-term renal function would be expected to improve after TAVR. However, our data, as well as previous data, do not support this assumption. In the study by De Backer et al., renal function did not improve one year after the procedure [13]. Also, in a study by Keles et al., the improvement observed in the first 30 days after the procedure disappeared at 6 months [30].

Renal function and hemoglobin level within three months of TAVR may be influenced by hemodynamic changes that occur during the procedure, blood loss, the loading of fluids or blood products, and changes in baseline medications. Therefore, data within this timeframe may not reflect the true effect of TAVR on renal function. Longer-term data may be influenced by aging and direct effects of comorbidities such as hypertension and diabetes. General deterioration over time would be expected for the patients in our cohort; thus, any improvement may be attributed to TAVR. We therefore believe that the time frame we used in our study, of 5–12 months after TAVR, represents the most accurate data on the direct influence of the procedure on renal function and hemoglobin levels.

## 5. Study Limitations and Generalizability

The main limitation of our study is its observational nature. While this is a single-center study, our cohort was relatively large. Notably, our data were collected during a phase in which the TAVR procedure had reached technological maturation and local expertise was established; thus, the data are up-to-date and may be generalizable to other populations. Importantly, our observations regarding the increased hemoglobin levels were still valid when we separately analyzed patients of a different ethnicity in our cohort [31].

## 6. Conclusions

In patients who underwent TAVR, a significant increase in hemoglobin was observed at 5–12 months after the procedure despite a reduction in renal function. Our findings imply that the gastrointestinal blood loss that commonly occurs in AS is significantly reduced 5–12 months after TAVR.

## Figures and Tables

**Table 1 jcm-13-05606-t001:** Baseline characteristics (*n* = 315).

Age	80.2 ± 7.0
Female (%)	168 (53%)
BMI	28.0 ± 5.6
NYHA ≥ III	216 (69%)
STS mortality, mean (*n* = 168)—%	3.8 ± 2.3
EuroScore 2, mean (*n* = 315)—%	5.5 ± 4.3
Diabetes mellitus	120 (38%)
Dyslipidemia	264 (84%)
Hypertension	278(88%)
Dialysis	0 (0%)
Atrial fibrillation	91 (29%)
CAD	158 (50%)
Prior cardiac surgery	52 (16%)
Previous stroke	19 (6%)
COPD	18 (6%)
Never smoked	242 (86%)
PAD	19 (6%)
Ejection fraction (%)	58 ± 9
Mean aortic valve gradient (mmHg)	47 ± 12

BMI = body mass index, NYHA = New York Heart Association, STS = Society of Thoracic Surgery, CAD = coronary artery disease, COPD = chronic obstructive pulmonary disease, PAD = peripheral artery disease.

**Table 2 jcm-13-05606-t002:** Change in hemoglobin (HB) in subgroups of patients after transcatheter aortic valve replacement (*n* = 315).

	HB			
	Before	After	*p* Value within Groups	*p* Value between Groups
	Mean ± standard deviation			
HB, *n* = 315	11.79 ± 1.65	12.18 ± 1.64	<0.0001	
Anemia				
No, *n* = 105	13.50 ± 1.00	13.20 ± 1.43	0.025	<0.0001
Yes, *n* = 210	10.94 ± 1.20	11.65 ± 1.49	<0.0001	
Gender				
male, *n* = 149	12.29 ± 1.72	12.60 ± 1.72	0.012	0.34
female, *n* = 174	11.36 ± 1.47	11.80 ± 1.46	<0.0001	
Valve_type:				
Corevalve, *n* = 3	12.67 ± 1.20	11.73 ± 2.90		0.055
EvoluteR, *n* = 182	11.80 ± 1.70	12.07 ± 1.60	0.020	
Evolut Pro, *n* = 48	11.80 ± 1.67	12.30 ± 1.64	0.010	
Sapien3, *n* = 82	11.66 ± 1.60	12.30 ± 1.65	<0.0001	
Diabetes mellitus				
No, *n* = 195	12.02 ± 1.60	12.32 ± 1.56	<0.0001	0.27
Yes, *n* = 120	11.42 ± 1.70	11.93 ± 1.72	0.001	
Hypertension				
No, *n* = 37	12.05 ± 1.60	12.52 ± 1.50	0.016	0.75
Yes, *n* = 278	11.72 ± 1.67	12.11 ± 1.64	<0.0001	

**Table 3 jcm-13-05606-t003:** Renal function before and after transcatheter aortic valve replacement.

	eGFR, *n* = 313			
	Before	After	*p* Value within Groups	*p* Value between Groups
	Mean ± Std			
	60.05 ± 24.1	58.30 ± 24.50	0.024	
Anemia				
No, *n* = 103	63.60 ± 21.33	61.35 ± 22.70	0.09	0.61
Yes, *n* = 210	58.32 ± 25.22	56.80 ± 25.24	0.11	
Gender				
male, *n* = 145	60.70 ± 23.02	59.00 ± 22.72	0.14	0.95
female, *n* = 168	59.52 ± 25.04	57.71 ± 26.00	0.09	
Valve type:				
Corevalve, *n* = 3	74.66 ± 43.30	74.40 ± 40.00	1.0	0.067
EvolutR, *n* = 182	60.90 ± 25.21	58.74 ± 24.20	0.047	
Evolut Pro, *n* = 48	55.85 ± 17.12	57.15 ± 21.05	0.48	
Sapien3, *n* = 82	60.10 ± 24.40	57.36 ± 26.68	0.053	
Diabetes mellitus				
No, *n* = 194	58.23 ± 22.36	56.64 ± 22.71	0.08	0.78
Yes, *n* = 119	63.02 ± 26.51	61.00 ± 27.04	0.15	
Hypertension				
No, *n* = 36	64.17 ± 24.47	62.58 ± 24.51	0.30	0.70
Yes, *n* = 277	59.38 ± 24.01	57.74 ± 24.48	0.045	

eGFR = estimated glomerular filtration rate.

**Table 4 jcm-13-05606-t004:** Change in hemoglobin before and after TAVR, univariable model.

	Mean ± Standard Deviation (Median)	*p* Value
Gender:		
Male, *n* = 147	0.33 ± 1.55 (0.20)	0.478
Female, *n* = 168	0.44 ± 1.34 (0.50)	
Diabetes mellitus		
No, *n* = 195	0.31 ± 1.29 (0.40)	0.218
Yes, *n* = 120	0.52 ± 1.65 (0.50)	
Hypertension		
No, *n* = 37	0.46 ± 1.17 (0.20)	0.738
Yes, *n* = 278	0.38 ± 1.47 (0.45)	
Correlation		
	R	*p* value
Age	0.08	0.891
eGFR, baseline	−0.09	0.115
eGFR, change (6-month baseline)	0.07	0.229
EuroScore 2	0.09	0.112

eGFR = glomerular filtration rate.

**Table 5 jcm-13-05606-t005:** Multivariable linear regression model for predicting a change in the hemoglobin level.

Variable	B	Standard Error	*p* Value	95% CI
Diabetes mellitus	0.241	0.174	0.166	−0.101, 0.583
Hypertension	−0.192	0.258	0.457	−0.70, 0.316
eGFR, baseline	−0.003	0.004	0.345	−0.01, 0.004
eGFR, change (6 months-baseline)	0.005	0.006	0.392	−0.007, 0.017
EuroScore 2	0.025	0.018	0.161	−0.010, 0.061

eGFR = glomerular filtration rate.

## Data Availability

The original study data are stored in a local station and are available on request to eitancardio@gmail.com.

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
