# Peer review of "Reduced Rate of Anemia after Transcatheter Aortic Valve Replacement"

_jcm, 2024, doi:10.3390/jcm13185606_

Round 1

Reviewer 1 Report

Comments and Suggestions for Authors

The abstract of the TAVR study requires further refinement. To strengthen its impact, additional details regarding transcatheter aortic valve replacement (TAVR) procedures should be included. Specifically, the claim that gastrointestinal (GI) blood loss is reduced post-TAVR needs substantiation through relevant literature citations. While the 1975 Declaration of Helsinki need not be explicitly mentioned, it’s advisable to focus on research methodology and statistical analysis. Elaborate on the prospective institutional TAVR registry, providing insights into patient selection criteria, data collection methods, and follow-up procedures. Additionally, consider comparing the study findings with national registries to establish context and baseline values. For a more nuanced analysis, explore bleeding outcomes based on the type of TAVR device used or the manufacturer. Overall, enhancing these aspects will contribute to a more robust and informative flow of this study.

Author Response

Comment 1: The abstract of the TAVR study requires further refinement

Response: As abstract length is limited, it includes only the data we conceder most important.

Comment 2:   To strengthen its impact, additional details regarding transcatheter aortic valve replacement (TAVR) procedures should be included. 

Response: TAVR is a well accepted procedure. We add a sentence in introduction that TAVR is usually performed via transfemoral approach.

Comment 3: the claim that gastrointestinal (GI) blood loss is reduced post-TAVR needs substantiation through relevant literature citations

Response: Thus far there is scarce data (citations 5-8). Our study aims to support this assumption.

 Comment 4: consider comparing the study findings with national registries to establish context and baseline values

Response: we compare our results with previous studies (citations 22-27)

Comment 5: For a more nuanced analysis, explore bleeding outcomes based on the type of TAVR device used or the manufacturer.

Response: Our data is large enough for general analysis but yet sub grouping it will result in too small numbers, loosing statistical power.

Reviewer 2 Report

Comments and Suggestions for Authors

Dear author ,

I would like to extend my sincere gratitude for your insightful research titled "Reduced Rate of Anemia after Transcatheter Aortic Valve Replacement." Your work contributes significantly to our understanding of the effects of TAVR on hemoglobin levels and renal function, especially among the elderly population with severe aortic stenosis.

I have some Suggestions for Improvement:

- In material and methods, please add the study ,design area (place ) and duration because this are very important for the research database. The study duration it mentioned in the result but it also important to mention same in the material and method.

- In discussion part of you add more further exploration of the mechanisms underlying the observed changes in hemoglobin and renal function would enhance the discussion.

- Also here are some notable gaps you can addressing these gaps in research could enhance the understanding of the effects of TAVR and help guide clinical practice for managing anemia and renal function in patients with severe aortic stenosis. If no way to addressing them now you add them to the part of the study limitations

- Lack of Control Group: The study does not include a control group of patients who did not undergo TAVR, making it challenging to attribute the changes in hemoglobin levels and renal function specifically to the TAVR procedure. A comparative analysis would strengthen the findings.

- Limited Follow-Up Duration: While the study assesses outcomes at 5-12 months post-TAVR, longer-term data (beyond 12 months) would provide more insights into the sustainability of anemia improvement and renal function changes over time.

- Patient Demographics and Comorbidities: The study does not delve deeply into how various demographic factors (age, sex, ethnicity) and comorbid conditions (diabetes, hypertension) might influence hemoglobin levels and renal function after TAVR. This lack of stratification limits the understanding of how different patient profiles might respond to the intervention.

- Assessment of Quality of Life: The paper does not discuss changes in the quality of life or functional status of the patients after TAVR. Such considerations are essential for understanding the broader impact of the procedure beyond clinical parameters.

- Effect of Medications: There is no mention of the medications taken by patients post-TAVR, such as antiplatelet therapy or iron supplementation. These could significantly affect hemoglobin levels and renal function and should be considered as potential confounders.

Author Response

Comment 1: study duration it mentioned in the result but it also important to mention same in the material and method.

Response: As detailed in Methods section, we collected Hemoglobin and Creatinine levels 5-12 months after the procedures. The same is detailed in Results  section.

Comment 2: Lack of Control Group

Response: As an observational study there is no control group but rather the observed change in the patients group after intervention. Patients not under-going TAVR these days are too sick patients - too fragile with high background illnesses and short life expectation. Thus there are no patients for comparison. We refer to this aspect in Limitations section.

Comment 3: Patient Demographics and Comorbidities, The study does not delve deeply into how various demographic factors (age, sex, ethnicity) and comorbid conditions (diabetes, hypertension) might influence hemoglobin levels and renal function after TAVR

Response: Patient Demographics and Comorbidities are detailed in table 1. Correlation between baseline characteristics and change in hemoglobin is presented in tables 4-5 and in Results section.

Comment 4: - Effect of Medications: There is no mention of the medications taken by patients post-TAVR, 

Response: we added a sentence in Results section: Patients after TAVR with no other anti-thrombotic indication are treated with Aspirin for life long. At the referred years the practice at our center was to add Clopidogrel for 3-6 months.